# Modifiable Risk Factor Possession Patterns of Dementia in Elderly with MCI: A 4-Year Repeated Measures Study

**DOI:** 10.3390/jcm9041076

**Published:** 2020-04-10

**Authors:** Osamu Katayama, Sangyoon Lee, Seongryu Bae, Keitaro Makino, Yohei Shinkai, Ippei Chiba, Kenji Harada, Hiroyuki Shimada

**Affiliations:** Department of Preventive Gerontology, Center for Gerontology and Social Science, National Center for Geriatrics and Gerontology, 7-430 Morioka-cho, Obu City 474-8511, Aichi, Japan; sylee@ncgg.go.jp (S.L.); bae-sr@ncgg.go.jp (S.B.); kmakino@ncgg.go.jp (K.M.); yshinkai@ncgg.go.jp (Y.S.); ichiba@ncgg.go.jp (I.C.); harada-k@ncgg.go.jp (K.H.); shimada@ncgg.go.jp (H.S.)

**Keywords:** mild cognitive impairment, latent class analysis, modifiable risk factor, psychosocial, health behavior, educational, smoking

## Abstract

This study clarified the patterns of possessing modifiable risk factors of dementia that can be corrected by the elderly who were primarily determined to have mild cognitive impairment (MCI), and then determined the relationship between retention patterns and outcomes from MCI through a 4-year follow-up study. The participants were 789 community-dwelling elders who were ≥65 years old with MCI at baseline. After 4 years, participants were classified into reverters and nonreverters, according to their cognitive function. Repeated measures analysis was performed after imputing missing values due to dropout. Nine modifiable risk factors at baseline were classified by latent class analysis. Subsequently, we performed binomial logistic regression analysis. The reversion rate of 789 participants was 30.9%. The possession patterns of modifiable risk factors among the elderly with MCI were classified into five patterns: low risk, psychosocial, health behavior, educational, and smoking factors. According to logistic regression analysis, the low risk factors class was more likely to recover from MCI to normal cognitive than the other classes (*p* < 0.05). These results may provide useful information for designing interventions to prevent cognitive decline and dementia in individuals with MCI.

## 1. Introduction

Mild cognitive impairment (MCI) is the intermediate state of cognitive function between the changes seen in aging, dementia, and often Alzheimer’s disease (AD) [1]. MCI is classified into two subtypes: amnestic and non-amnestic [2]. Amnestic MCI is clinically significant memory impairment that does not meet the criteria for dementia. Typically, patients and their families are aware of the increasing forgetfulness. However, other cognitive capacities, such as executive function, use of language, and visuospatial skills, are relatively preserved, and functional activities are intact, except perhaps for some mild inefficiencies. On the other hand, non-amnestic MCI is characterized by a subtle decline in the functions not related to memory, affecting attention, use of language, or visuospatial skills [1]. Individuals with MCI, or those who experience impairment across one or more cognitive domains (e.g., memory, language, thinking, and judgment), beyond that expected for age, are at an increased risk of progressing to AD and other types of dementias [3,4,5]. This is concerning, given that up to 20% of people over the age of 65 experience MCI [6,7]. Various factors have been related to the onset of dementia. According to Livingston et al. (2017), approximately 65% are genetic factors, such as apolipoprotein E (APOE), and other nonmodifiable risk factors. The remaining 35% report nine modifiable risk factors, such as less education, hearing loss, hypertension, obesity, smoking, depression, physical inactivity, social isolation, and diabetes [8]. A meta-analysis on dementia outcomes revealed a pooled relative risk for dementia of 1.20 for one risk factor, 1.65 for two, and 2.21 for three or more, relative to no risk factors. For dementia prevention, modifiable risk factor exposure should be kept to a minimum, and exposure to additional modifiable risks should be avoided [9]. Cognitive function may be improved by simultaneous intervention for multidomain lifestyles, such as dietary counseling, exercise, and cognitive training, and by vascular risk factor monitoring [10,11].

Additionally, in previous MCI longitudinal studies, many MCI cases did not develop dementia but reverted to normal cognition (NC) [12,13]. Cognitive function, demographics, genetic data, personality, and lifestyle activities have been reported as factors associated with recovery from MCI to NC [13,14,15,16,17]. A meta-analysis reveals that factors such as young age, high educational level, and Mini-Mental State Examination (MMSE) score are more likely to revert from MCI to NC without APOEe4 allele, hypertension, and stroke.

However, the relationship between the pattern of possessing modifiable risk factors for elderly people with MCI and the outcome remains unclear. If MCI is the intermediate state of cognitive function between the changes observed in aging, dementia, and often AD, clarifying the relationship between the possession pattern of modifiable risk factors and the outcome may contribute to the suppression of dementia or AD onset. Therefore, in this study, we aim to clarify the possession pattern of modifiable risk factors that can be corrected by the elderly who were first determined to have MCI. We also elucidate the relationship between retention patterns and outcomes from MCI through a 4-year follow-up.

## 2. Experimental Section

### 2.1. Study Participants, Design, and Setting

We selected our participants from adults enrolled in a population-based cohort study entitled “The Obu Study of Health Promotion for the Elderly (OSHPE)” [18], which is part of the NCGG-SGS [19]. Then, we analyzed the longitudinal data from 461 community-dwelling elders who were ≥65 years old (mean age: 70.9 ± 4.5 years, 223 men and 238 women), had joined both the first and second waves of the OSHPE, and had MCI during the first wave assessment. During the first wave, which was held between August 2011 and February 2012, 5104 community-dwelling elderly people participated in a baseline OSHPE assessment. In the second-wave cognitive examination held between August 2015 and August 2016, 3095 (60.6%) remained to participate. The inclusion criteria were residence in Obu City and ≥65 years of age during the first examination (August 2011 to March 2013). Conversely, the baseline exclusion criteria were as follows: health problems, such as AD, Parkinson’s disease, or stroke (*n* = 312); inability to perform basic daily life activities, such as eating, grooming, bathing, and climbing up and down stairs (*n* = 22); need for support or care, as certified by the Japanese public long-term care insurance system, due to disability (*n* = 110); inability to complete cognitive tests at the baseline assessment (*n* = 159); relocation (*n* = 30) or death (*n* = 85) during the follow-up period; NC (*n* = 3026); and global cognitive decline at the baseline assessment (*n* = 571). Of the 789 potential participants, 328 did not receive the second-wave cognitive examination (Figure 1). Thus, we analyzed data from 461 participants. Multivariate normal imputation was used to adjust for selection bias and loss of information, because we found potential bias of the sample in baseline data on those who remained versus those who left during the follow-up (Appendix A). In this study, we imputed reversion status, which was divided into reverters (those who recovered from MCI to NC) and non-reverters (those who had MCI, global cognitive impairment (GCI) (as indicated by an MMSE [20] score of < 24), and/or AD) for participants with missing data. We generated imputed values for each participant with missing data using a multivariate normal imputation procedure assuming that those values were missing at random [21]. Generally, the multivariate normal imputation utility imputes missing values based on the multivariate normal distribution. The algorithm uses the least squares imputation. In addition, pairwise covariance is used to construct the covariance matrix. The diagonal entries (variances) are computed using all nonmissing values for each variable. The off-diagonal entries for any two variables are computed using all observations that are present for both variables. If the covariance matrix is singular, the algorithm uses the minimum norm least squares imputation, based on the Moore-Penrose pseudo-inverse. The multivariate normal imputation allows the option to employ a shrinkage estimator for the covariances. The use of shrinkage estimators can improve the estimation of the covariance matrix [21]. Each participant provided informed consent before being included in the study. The study protocol was approved by an institutional review board.

### 2.2. Measurements of Modifiable Risk Factors

Participants completed a questionnaire that contained questions regarding less education, hearing loss, hypertension, smoking, physical inactivity, social isolation, and diabetes as various elements of modifiable risk factors. As for medical information (e.g., hearing loss, hypertension, and diabetes) in the questionnaire items, a staff member qualified as a nurse interviewed the participants face-to-face. Other items were carefully evaluated face-to-face by our trained staff. In measuring education, the questionnaire included the question (1) “How many years did you go to school?” Education of ≤ 10 years indicated a risk factor. Items that measured hearing loss included the question (2) “Can you hear normally?” A “No” answer indicated a risk factor. Items that measured hypertension included the question (3) “Have you ever been diagnosed with hypertension?” A “Yes” answer indicated a risk factor. Items that measured smoking included the question (4) “Are you currently smoking?” A “Yes” answer indicated a risk factor. Items that measured physical inactivity included the questions (5) “How many days a week do you engage in low-intensity physical exercise?” and (6) “How many days a week do you engage in moderate-intensity physical exercise or sports?” If neither was done (i.e., no physical activity), we indicated a risk factor. Items that measured social isolation included the question (7) “Do you often go out to visit your friends?,” (8) “Do you ever think you are supportive to your friends and family?,” and (9) “Do you talk to someone every day?” A “No” to any one of the three answers indicated a risk factor. Additionally, (10) “Compared with last year, do you go out less often?” A “Yes” answer indicated a risk factor. Items that measured diabetes included the question (11) “Have you ever been diagnosed with diabetes?” A “Yes” answer indicated a risk factor. Obesity, defined as “BMI ≥ 25”, was regarded as a risk factor. Depressive symptoms were measured using the 15-item Geriatric Depression Scale (GDS) [22]. We defined “GDS ≥ 5” as a risk factor.

### 2.3. Measurement of Cognitive Functions and Incident AD

To conduct cognitive screening, we used the National Center for Geriatrics and Gerontology-Functional Assessment Tool (NCGG-FAT), which is an iPad application [23]. The NCGG-FAT includes the following domains: (1) memory (word list memory-I [immediate recognition] and word list memory-II [delayed recall]); (2) attention (a tablet version of the Trail Making Test-part A; TMT-part A); (3) executive function (a tablet version of the TMT-part B), and (4) processing speed (a tablet version of the Digit Symbol Substitution Test). The NCGG-FAT has a high test–retest reliability and a moderate to high criterion-related validity [23] and predictive validity [24] among community-dwelling elders. Cognitive assessments were conducted by staff who were trained by the authors. We identified potential participants with MCI after reviewing the available clinical, neuropsychological, and laboratory data at meetings involving study neurologists and neuropsychologists, as previously described [18]. In brief, participants with MCI were independently recruited using the NCGG-FAT, which has two memory tasks, attention and executive function tests, and a processing speed task. Using the established criteria [2], we diagnosed MCI in individuals who exhibited cognitive impairment but were functionally independent in terms of basic daily life activities. For all cognitive tests, we used the established standardized thresholds in each corresponding domain to define impairment in population-based cohorts comprising community-dwelling elders (scores of >1.5 standard deviations [SDs] that specified the age and educational means). Global cognitive function was measured using the MMSE [25]. Specifically, we used <24 points on the MMSE as the cutoff score for GCI, conforming to previous findings [26]. Participants whose cognitive test scores were all >1.5 SD units above the mean belonged to the NC group. In the present study, participants were tracked monthly for newly incident AD, as recorded by the Japanese National Health Insurance and Later-Stage Medical Care systems [27]. Participants had AD if they had been diagnosed by a medical doctor according to the International Classification of Diseases, 10th revision. Incidence of AD was based simply on diagnoses by third-party doctors who were blinded to the design and participant groups of the study. Participants were divided into two groups as follows: reverter group, who had NC during follow-up, and nonreverter group, who had MCI, GCI, and/or AD during follow-up.

### 2.4. Statistical Analysis

We calculated the reversion rates from MCI to NC during the follow-up assessments. Differences in the baseline participant characteristics and differences between the reverter and nonreverter groups were examined by unpaired *t*-test and Pearson chi-squared test, respectively. Residuals followed the t distribution wherein *t* > 1.96 indicates *p* < 0.05 and *t* > 2.56 indicates *p* < 0.01.

For the clustering of modifiable risk factor data, we used latent class analysis (LCA) (Latent Gold V.5.1). In Latent Gold, latent class models are estimated with a full information maximum likelihood or maximum posterior mode algorithm, which uses all the available information for each individual to compute parameter estimates. In our study, the optimal number of clusters were identified using the Bayesian Information Criterion (BIC), Akaike Information Criterion (AIC), and AIC with a penalty factor of 3 (AIC3) [28,29]. Smaller values of each indicate a better fit. Subsequently, we performed binomial logistic regression analysis, with reversion status as the dependent variable (the most typical, nonreverters as a reference group) and cluster membership as the independent variable. First, we used unadjusted models. Second, for cluster membership, we adjusted for covariates such as age, sex, education history, heart disease, pulmonary disease, MMSE, and GDS. Data are presented as ORs with 95% CI. The significance level was set at *p* < 0.05. All analyses were performed using the IBM SPSS version 25.0 (IBM Japan, Tokyo).

## 3. Results

### 3.1. Baseline Characteristics of the Participants

In the initial group of 461 participants, 234 (50.8%) reverted from MCI to NC. The reversion rate of participants for whom we imputed samples was 30.9%. Table 1 lists the participant baseline characteristics for those grouped according to the change patterns from MCI in the imputed group. Age, educational level, GDS score, category of MCI, and modifiable risk factors differed significantly between reverters and nonreverters (*p* < 0.05).

### 3.2. LCA of Modifiable Risk Factor Possession Patterns

The fit criteria for the LCA on 789 participants are presented in Table 2. When further latent class was added, the AIC information criterion and likelihood continued to decline, whereas BIC and AIC3 were reversed. Meanwhile, AIC evaluation showed an improvement of goodness-of-fit with an increasing number of classes up to 5. The AIC and likelihood suggested that the five classes fitted well. Therefore, we chose a 5-class model. According to the prior probability shown in Table 3, we labeled the classes as psychosocial factors (Class 1; *n* = 186, 25.9%), health behavior factors (Class 2; *n* = 179, 23.9%), low risk factors (Class 3; *n* = 159, 20.3%), educational factor (Class 4; *n* = 146, 18.5%), and smoking factor (Class 5; *n* = 199, 11.4%).

### 3.3. Demographical and Modifiable Risk Factor Cluster Characteristics

Table 4 shows the raw baseline data of demographics, education history, underlying disease, walking speed, MMSE, GDS, category, and modifiable risk factor characteristics by cluster. The class differences for age, sex, education, hypertension, walking speed, MMSE, GDS, and modifiable risk factors were observed. Table 4 also presents the post hoc differences.

Class 1 (psychosocial factors) was characterized by older age, lower education level, slower walking speed, and having modifiable risk factors related to psychosocial aspects such as high depression tendency and social isolation (*p* < 0.05). Class 2 (health behavior factors) was characterized by younger age, male dominance, longer education level, and having modifiable risk factors, such as hypertension, obesity, and smoking, which are related to health behaviors (*p* < 0.05). Class 3 (low risk factors) was characterized by younger age, female dominance, longer education level, faster walking speed, higher MMSE, lower GDS, and having fewer modifiable risk factors (*p* < 0.05). Class 4 (educational factors) was characterized by older age, female dominance, lower educational level, slower walking speed, and having lower MMSE (*p* < 0.05). Class 5 (smoking factors) was characterized by male dominance, faster walking speed, and having a smoking habit (*p* < 0.05).

Table 5 shows the ORs and 95% CIs estimated by unadjusted and adjusted binomial logistic regression analyses, with reversion status as the dependent variable (with the nonreverter group as reference) and the LCA class as the independent variable. After adjusting for age, sex, education history, heart disease, pulmonary disease, MMSE, and GDS, class 1 (psychosocial factors, OR: 0.46; 95% CI: 0.23–0.92), class 2 (health behavior factors, OR: 0.42; 95% CI: 0.26–0.70), class 4 (educational factors, OR: 0.51; 95% CI: 0.27–0.94), and class 5 (smoking factors, OR: 0.34; 95% CI: 0.18–0.63) had significantly lower odds of being reverters.

## 4. Discussion

In this study, we clarified the possession pattern of modifiable risk factors that can be corrected by elders who were first determined to have MCI. We also clarified the relationship between retention patterns and outcomes from MCI through a 4-year follow-up. The MCI to NC reversion rate ranges from 10% to 50%, depending on the follow-up period and MCI subtype [6,7]. In a previous study with a study population similar to ours and follow-up, the reversion rate was 35% [30], which is similar to the 30.9% rate in our study.

According to our study results, community-dwelling elders with MCI showed five patterns of modifiable risk factor possession. At the baseline, the characteristics of demographics, education history, underlying disease, walking speed, MMSE, GDS, MCI category, and modifiable risk factors of each class were different. Most participants (25.9%) belonged to the psychosocial factors class, which has factors such as depression and social isolation and is considered to be strongly related to late life [8]. Furthermore, psychosocial factors class was significantly less associated with reverters, compared with the low risk factors class. The reason may be that depression and social isolation are related to cognitive decline and dementia [31,32]. Recent systematic reviews and meta-analyses have also shown that depression increases the risk of dementia [33]. Furthermore, a systematic review and meta-analysis of social isolation and cognitive function revealed that social isolation was associated with cognitive function in later life [34]. Meanwhile, the health behavior factors class accounted for 23.9% of the total participants. They had health behavior factors such as hypertension, obesity, and smoking. This class appeared to be a class with factors related to midlife [8]. A previous study has reported that high BMI is associated with the incidence of hypertension and diabetes. Smoking also had a direct effect on the incidence of diabetes [35]. Hypertension, obesity, and smoking are also associated with cognitive decline and dementia [36,37,38]. Therefore, this class is likely to be significantly less associated with reverters than the low risk factors class. The educational factors class and smoking factors class accounted for 18.5% and 11.4% of the total participants, respectively. A low educational level characterized the educational factors class, which was considered to be related to early life [8]. In addition, a low educational level is associated with impaired cognitive function and dementia [39]. Meanwhile, the smoking factors class was characterized by a risk factor of smoking. There were significantly more men in this class. In 2011, the rate of smoking among Japanese adults was 29.3% for men and 6.4% for women for those aged 60–69 years. For those aged 70 and older, 16.6% were male and 3.0% were female, indicating that the rate of smoking is clearly higher in men [40]. Smoking is clearly a factor associated with cognitive decline and dementia [41]. In addition, smoking is a common negative factor for adherence to the multidomain intervention of the Finnish Geriatric Intervention Study to Prevent Cognitive Impairment and Disability and the Multidomain Alzheimer Preventive Trial [11,42,43]. Smoking is also a factor that causes hypertension. However, Class 4 did not include participants with hypertension. Hypertension has been shown to be more prevalent in women [44,45,46]. Therefore, it may be related to the high number of male participants in this class. From these factors, both educational factors class and smoking factors class are likely to be significantly less associated with reverters than the low risk factors class. Hence, elders with MCI have multiple patterns of modifiable risk factor possession. Overall, as mentioned, elders with MCI have several patterns of possessing modifiable risk factors. In addition, possessing modifiable risk factors was negatively related to the reversion from MCI to NC.

However, our study has limitations. For instance, the 50.8% reversion rate from MCI to NC before sample imputation may have been affected by false positives; thus, the influence of modifiable risk factor possession patterns on recovery from MCI to NC is difficult to accurately determine. Moreover, participants were not randomly recruited, and approximately 42% of the participants were lost to follow-up. This rate may have led to the underestimation of cognitive decline. Furthermore, given that it did not cover all modifiable risk factors, it is not possible to consider the effects of other modifiable risk factors such as sleep disturbances and eating habits. In addition, the involvement of genetic factors, which are not modifiable, has not been examined. Nevertheless, this study had the following strengths and implications. Our findings are consistent with those of comprehensive geriatric assessments designed to collect information on modifiable risk factors. To the best of our knowledge, this study is the first to classify modifiable risk factor possession patterns using LCA and clarify their relationship with MCI status. Our results provide additional information to further understand the possession patterns of modifiable risk factors in elders with MCI and implement appropriate interventions for each individual.

## 5. Conclusions

This study formulated five possession patterns (classes) of modifiable risk factors among elders with MCI. The low risk factors class is more likely to recover from MCI to NC than the other classes. These results may provide useful information for designing interventions to prevent cognitive decline and dementia in individuals with MCI.

## Figures and Tables

**Figure 1 jcm-09-01076-f001:**
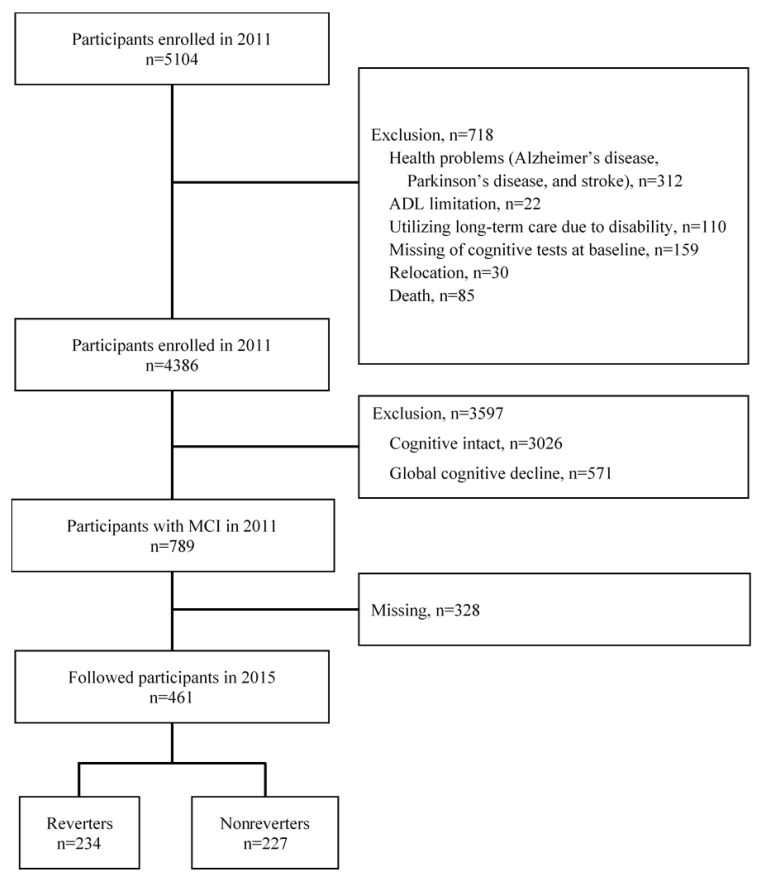
Flow diagram of sample selection.

**Table 1 jcm-09-01076-t001:** Comparison of baseline characteristics between reverters and nonreverters.

	Reverters (*n* = 244)	Nonreverters (*n* = 545)	*p* Value
Age (years) *	69.5 (3.4)	73.1 (5.7)	<0.01 ^a^
Sex (% male)	41.8	48.8	>0.05 ^b^
Education (years) *	11.8 (2.4)	11.0 (2.4)	<0.01 ^a^
Heart disease (% yes)	16.4	17.4	>0.05 ^b^
Pulmonary disease (% yes)	7.4	10.1	>0.05 ^b^
Hypertension (% yes)	8.2	10.4	>0.05 ^b^
Walking speed (m/s) *	1.3 (0.2)	1.2 (0.2)	>0.05 ^a^
Mini-mental state examination (points) *	26.9(1.9)	26.4(1.8)	>0.05 ^a^
Geriatric depression scale (points) *	2.6 (2.3)	3.3 (2.8)	<0.01 ^a^
**Category of MCI (%)**			<0.01 ^b^
amnestic MCI single domain	16.4	13.2	
non-amnestic MCI single domain	72.1 ^c^	57.4 ^d^	
amnestic MCI multiple domain	4.5 ^d^	8.8 ^c^	
non-amnestic MCI multiple domain	7.0 ^d^	20.6 ^c^	
**Modifiable risk factors (% yes)**			
Less education (10 years ≥)	24.2	40.2	<0.01 ^b^
Hearing loss	3.0	4.3	>0.05 ^b^
Hypertension	41.4	50.3	<0.05 ^b^
Obesity (BMI 25≤)	25.5	29.0	>0.05 ^b^
Smoking	34.4	42.6	<0.05 ^b^
Depression (GDS 5≤)	16.8	26.3	<0.01 ^b^
Physical inactivity	22.5	34.3	<0.01 ^b^
Social isolation	27.5	40.7	<0.01 ^b^
Diabetes	12.3	16.0	>0.05 ^b^

* Mean (Standard Deviation). ^a^
*p* values reported from unpaired *t*-test. ^b^
*p* values obtained by Pearson’s chi square test. ^c^ Statistically significant association by adjusted standardized residual >1.96 (*p* < 0.05). ^d^ Statistically significant association by adjusted standardized residual <−1.96 (*p* < 0.05).

**Table 2 jcm-09-01076-t002:** Parameters of fit in latent class analysis.

Classes (N)	Parameters (N)	Likelihood	BIC	AIC	AIC3
3	29	−3865.8	7925.0	7789.5	7818.5
4	39	−3852.3	7964.8	7782.6	7821.6
5	49	−3839.6	8006.0	7777.2	7826.2
6	59	−3832.5	8058.6	7783.0	7842.0

BIC, Bayesian Information Criterion; AIC, Akaike Information Criterion; AIC3, Akaike Information Criterion with penalty factor of 3.

**Table 3 jcm-09-01076-t003:** Conditional probabilities of latent modifiable risk factor classes.

	Class 1 (Psychosocial Factors)	Class 2 (Health Behavior Factors)	Class 3 (Low Risk Factors)	Class 4 (Educational Factor)	Class 5 (Smoking Factor)
Class prevalence (%)	25.9	23.9	20.3	18.5	11.4
Less education (10 years ≥)	0.4547	0.1676	0.0331	0.8196	0.3203
Hearing loss	0.0501	0.0485	0.0255	0.0322	0.0255
Hypertension	0.4638	0.8573	0.2521	0.5248	0.0155
Obesity (BMI 25≤)	0.2782	0.5323	0.0768	0.2764	0.1026
Smoking	0.4258	0.5957	0.1713	0.0135	0.9713
Depression (GDS 5≤)	0.7428	0.0741	0.0823	0.0298	0.0097
Physical inactivity	0.4749	0.2936	0.0695	0.2915	0.4015
Social isolation	0.7365	0.2378	0.167	0.3016	0.2667
Diabetes	0.1635	0.2361	0.0759	0.0983	0.1434

**Table 4 jcm-09-01076-t004:** Comparison of baseline characteristics of each class.

	Psychosocial Factors *n* = 186 (23.6%)	Health Behavior Factors *n* = 179 (22.7%)	Low Risk Factors *n* = 159 (20.2%)	Educational Factors *n* = 146 (18.5%)	Smoking Factorsn *n* = 119 (15.1%)	*p* Value	Post Hoc
Age (years) *	73.3 (5.9)	71.1 (4.4)	70.35 (4.2)	73.7 (5.9)	71.4 (5.5)	<0.05 ^a^	1,4>3 1,4>2 1>5 4>5
Sex (% male)	45.2	68.2 ^c^	21.4 ^d^	13.0 ^d^	91.6 ^c^	<0.01 ^b^	
Education (years) *	10.6 (2.2)	12.5 (2.2)	12.7 (1.9)	8.6 (0.9)	11.7 (2.3)	<0.05 ^a^	3>1,4,5 2>1,4,5 1>4 5>1 5>4
Heart disease (% yes)	19.9	21.2	13.8	16.4	11.8	>0.05 ^b^	
Pulmonary disease (% yes)	8.1	10.6	7.5	8.9	11.8	>0.05 ^b^	
Hypertension (% yes)	48.9	92.2 ^c^	26.4 ^d^	52.7	0.0 ^d^	<0.01 ^b^	
Walking speed (m/s) *	1.2 (0.2)	1.2 (0.2)	1.4 (0.2)	1.2 (0.2)	1.3 (0.2)	<0.05 ^a^	3>1,2,4,5 2>1 5>1 5>4
Mini-mental state examination (points) *	26.4 (1.8)	26.7 (1.8)	27.1 (1.9)	26.1 (1.7)	26.4 (1.8)	<0.05 ^a^	3>1,4,5 2>4
Geriatric depression scale (points) *	6.7 (2.4)	2.0 (1.5)	1.8 (1.6)	2.3 (1.3)	1.8 (1.5)	<0.05 ^a^	1>2,3,4,5
**Category of MCI**						>0.05 ^b^	
amnestic MCI single domain	11.3	19.6	13.8	8.2	18.5		
non-amnestic MCI single domain	59.7	57.5	68.6	64.4	60.5		
amnestic MCI multiple domain	10.8	5.6	4.4	8.9	7.6		
non-amnestic MCI multiple domain	18.3	17.3	13.2	18.5	13.4		
**Modifiable risk factors (% yes)**							
Less education (10 years ≥)	44.6 ^c^	8.9 ^d^	0.0 ^d^	100.0 ^c^	27.7	<0.01 ^b^	
Hearing loss	4.3	5.0	3.1	3.4	2.5	>0.05 ^b^	
Hypertension	48.9	92.2 ^c^	26.4 ^d^	52.7	0.0 ^d^	<0.01 ^b^	
Obesity (BMI 25≤)	26.9	58.1 ^c^	7.5 ^d^	30.1	7.6 ^d^	<0.01 ^b^	
Smoking	43.0	65.4 ^c^	0.0 ^d^	0.0 ^d^	100.0 ^c^	<0.01 ^b^	
Depression (GDS 5≤)	91.4 ^c^	3.4 ^d^	5.0 ^d^	0.0 ^d^	0.0 ^d^	<0.01 ^b^	
Physical inactivity	48.4 ^c^		5.0 ^d^	30.1	37.8	<0.01 ^b^	
Social isolation	73.1 ^c^		18.9 ^d^	33.6	27.7 ^d^	<0.01 ^b^	
Diabetes	15.6		5.7 ^d^	11.6	13.4	<0.01 ^b^	

* Mean (Standard Deviation) a *p* values reported from one-way ANOVA. Significant difference obtained by Games-Howell post-hoc test. ^b^
*p* values obtained by Pearson’s chi square test. ^c^ Statistically significant association by adjusted standardized residual > 1.96 (*p* < 0.05). ^d^ Statistically significant association by adjusted standardized residual < −1.96 (*p* < 0.05). 1, Psychosocial factors; 2, Health behavior factors; 3, Low risk factors; 4, Educational factors; 5, Smoking factors.

**Table 5 jcm-09-01076-t005:** Binomial logistic regression analysis with cluster membership as dependent variable.

	Unadjusted Models			Adjusted Models		
	Nonreverters (*n* = 545)	Reverters (*n* = 244)		Nonreverters (*n* = 545)	Reverters (*n* = 244)	
		OR (95% CI)	*p* value		OR (95% CI)	*p* value
Age (years)				REF	0.86 (0.83–0.90)	<0.01
Sex (male)				REF	0.93 (0.63–1.39)	>0.05
Education history (years)				REF	1.03 (0.95–1.13)	>0.05
Heart disease (yes)				REF	1.02 (0.65–1.60)	>0.05
Pulmonary disease (yes)				REF	0.84 (0.46–1.52)	>0.05
MMSE (points)				REF	1.08 (0.99–1.19)	>0.05
GDS (points)				REF	0.94 (0.85–1.04)	>0.05
Class						
Low risk factors	REF	1		REF	1	
Psychosocial factors	REF	0.25 (0.15–0.39)	<0.01	REF	0.46 (0.23–0.92)	<0.05
Health behavior factors	REF	0.39 (0.25–0.60)	<0.01	REF	0.42 (0.26–0.70)	<0.01
Educational factors	REF	0.29 (0.18–0.48)	<0.01	REF	0.51 (0.27–0.94)	<0.05
Smoking factors	REF	0.30 (0.18–0.51)	<0.01	REF	0.34 (0.18–0.63)	<0.01

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
