# Peer review of "Modifiable Risk Factor Possession Patterns of Dementia in Elderly with MCI: A 4-Year Repeated Measures Study"

_jcm, 2020, doi:10.3390/jcm9041076_

Round 1
Reviewer 1 Report
Dear authors,
This is a very interesting manuscript about the factors than can be noticeable in the relation between healthy aging and cognitive impairment in older adults. The results are very well explained, but there are some aspects that must be improved before its publication.
- Introduction: The introduction is, in my opinion, too short. The important aspects of the results in previous publications must be explained better, with more details, as well as including more articles, for example. A good definition of Mild Cognitive Impairment must be included.
- Objectives: The objectives, also at the end of the introduction section, must be modified. The objectives must be written in a future verbal tense, not in past tense (for example: "We clarified..." must be changed in "We aim to clarify...").
- Method section: this section is, in general, well defined and explained. However, I am concerned about the subsection called "Measurements of modifiable risk factors". This subsection includes the collection of very rellevant variables for the study. Nevertheless, some of this variables are collected with very simple questions, not with validated questionnaires about the topic. How the authors explain that? Is the validity of this variables compromised? How is possible to correct that?
- Results: good explanation, with important tables and figures.
- Disscussion: well discussed and explained, but more details in the introduction, as well as more articles, have to be included, and then also in this section.
- I would like to add a comment about the title. I think I would not use "longitudinal" in a 4-year study, I would prefer the term "repeated measures". Please, think about the characteristics of your study in terms of method and design.
Minnor comments: Line 75 - there are some words together that have to be separated.
Author Response
We thank the reviewer for fruitful suggestions, especially for suggesting better terms, sentences and helping me explaining my work more thoroughly. These are the answer to your questions.
To REVIEWER 1:
- Comments from the reviewer:
Introduction:
The introduction is, in my opinion, too short. The important aspects of the results in previous publications must be explained better, with more details, as well as including more articles, for example. A good definition of Mild Cognitive Impairment must be included.
Response:
I included the definition and clinical symptoms of Mild Cognitive Impairment in the Introduction, citing previous studies.
(line 32-42 of page 1)
- Comments from the reviewer:
Objectives:
The objectives, also at the end of the introduction section, must be modified. The objectives must be written in a future verbal tense, not in past tense (for example: "We clarified..." must be changed in "We aim to clarify...").
Response:
I changed the objectives from "We clarified" to "We aim to clarify". (line 63 of page 2)
In addition, I revised "We also elucidated" to "We also elucidate" written in Introduction. (line 65 of page 2)
- Comments from the reviewer:
Method section:
This section is, in general, well defined and explained. However, I am concerned about the subsection called "Measurements of modifiable risk factors". This subsection includes the collection of very rellevant variables for the study. Nevertheless, some of this variables are collected with very simple questions, not with validated questionnaires about the topic. How the authors explain that? Is the validity of this variables compromised? How is possible to correct that?
Response:
We also consider it a very important point. Therefore, the following contents have been added.
The questions were carefully evaluated face-to-face by our trained staff. In particular, medical information was evaluated by qualified nurses. (line 108-10 of page 3-4)
- Comments from the reviewer:
Results:
Good explanation, with important tables and figures.
Response:
Thank you very much for your comments.
- Comments from the reviewer:
Disscussion:
Well discussed and explained, but more details in the introduction, as well as more articles, have to be included, and then also in this section.
Response:
I have added detailed information on the modifiable risk factors mentioned in the Introduction, citing previous studies. (line 246-9, 251-5, 260-3, 266-9 of page 9)
- Comments from the reviewer:
I would like to add a comment about the title. I think I would not use "longitudinal" in a 4-year study, I would prefer the term "repeated measures". Please, think about the characteristics of your study in terms of method and design.
Response:
I changed the title from "A 4-year longitudinal study" to "A 4-year repeated measures study". (line 3-4 of page 1)
In addition, I revised "Longitudinal analysis" written in Abstract to "Repeated measures analysis". (line 18 of page 1)
- Comments from the reviewer:
Minnor comments:
Line 75 - there are some words together that have to be separated.
Response:
I separated those words. (line 85 of page 2)
Sincerely,
Osamu Katayama,
Department of Preventive Gerontology, Center for Gerontology and Social Science, National Center for Geriatrics and Gerontology, 7-430 Morioka-cho, Obu City, Aichi 474-8511, Japan.
Tel/Fax: +81-562-44-5651/+81-562-45-5638; E-mail: katayama.o@ncgg.go.jp

Reviewer 2 Report
This study investigated the relationship between the patterns of 9 modifiable risk factors and reversion to normal cognition (NC) or non-reversion in 789 community-dwelling elders with MCI (mean age: 70.9 ± 4.5 yr) over a 4-yr follow-up. Sample imputation was performed for analysis. The possession patterns of 9 modifiable risk factors for dementia (education, hearing, HBP, obesity, smoking, depression, physical inactivity, social isolation, DM) were classified into five classes: psychosocial, health behavior, low risk, educational, and smoking factors. Logistic regression analysis showed that low risk factors class at baseline were more associated with reversion from MCI to NC. The present study has some advantages including focus on modifiable risk factors and relatively large sample size.
Some comments are as follows:
Sentence 2 in page 2
-> Rewording would be recommended.
Line 75 of page 2: 396 participants
-> Please check it out again. Likewise for spacing.
Line 79 of page 2: GCI
-> For abbreviation of GCI, please use the word in full, initially.
Questionnaire (5) and (6) measuring physical inactivity in page 3: “How many days a week do you engage in low- or moderate-intensity physical exercise (or sports)?” If neither was done, we indicated a risk factor.
-> I wonder if only one day a week of low-intensity physical activity was rated as physical activity (i.e., no physical inactivity). Please make it clear.
Questionnaire (7)–(9) among items measuring social isolation in page 3:
-> Likewie, I wonder if social isolation means any “No” answer or all “No” answers for three items.
Line 127–8 of page 4: …two memory tasks (attention and executive function tests)..
-> Please recheck it.
Table 3: heading of columns
-> Please put the contents for Class 1–2, 4–5 as Class 3.
Table 4: number of each Class
-> Please recheck the number of each Class. It does not fit with number of total participants.
Table 4: Smoking factors Class
-> I wonder why Class 5 (Smoking factors) was characterized by no hypertension. Authors could mention these contents in Discussion.
Line 256–7 in page 9 (Discussion)
-> Rewording would be recommended.
Author Response
We thank the reviewer for fruitful suggestions, especially for suggesting better terms, sentences and helping me explaining my work more thoroughly. These are the answer to your questions.
To REVIEWER 2:
- Comments from the reviewer:
Sentence 2 in page 2
-> Rewording would be recommended.
Response:
I corrected the wording. (line 54-6 of page 2)
- Comments from the reviewer:
Line 75 of page 2: 396 participants
-> Please check it out again. Likewise for spacing.
Response:
I revised the number of participants from "396" to "461" and I inserted space. (line 85 of page 2)
- Comments from the reviewer:
Line 79 of page 2: GCI
-> For abbreviation of GCI, please use the word in full, initially.
Response:
I have added the full spelling of "GCI". (line 89-90 of page 2)
- Comments from the reviewer:
Questionnaire (5) and (6) measuring physical inactivity in page 3: “How many days a week do you engage in low- or moderate-intensity physical exercise (or sports)?” If neither was done, we indicated a risk factor.
-> I wonder if only one day a week of low-intensity physical activity was rated as physical activity (i.e., no physical inactivity). Please make it clear.
Response:
I added this explanation in the text. (line 119 of page 4)
- Comments from the reviewer:
Questionnaire (7)–(9) among items measuring social isolation in page 3:
-> Likewie, I wonder if social isolation means any “No” answer or all “No” answers for three items.
Response:
I added this explanation in the text. (line 122 of page 4)
- Comments from the reviewer:
Line 127–8 of page 4: …two memory tasks (attention and executive function tests)..
-> Please recheck it.
Response:
I checked and corrected. (line 139-41 of page 4)
- Comments from the reviewer:
Table 3: heading of columns
-> Please put the contents for Class 1–2, 4–5 as Class 3.
Response:
I put the contents for Class 1–2, 4–5 as Class 3. (Table 3)
- Comments from the reviewer:
Table 4: number of each Class
-> Please recheck the number of each Class. It does not fit with number of total participants.
Response:
I checked and corrected. (Table 4)
- Comments from the reviewer:
Table 4: Smoking factors Class
-> I wonder why Class 5 (Smoking factors) was characterized by no hypertension. Authors could mention these contents in Discussion.
Response:
I added this explanation to the Discussion. (line 260-3, 266-9 of page 9)
- Comments from the reviewer:
Line 256–7 in page 9 (Discussion)
-> Rewording would be recommended.
Response:
I corrected the wording. (line 280-1 of page 9)
Sincerely,
Osamu Katayama,
Department of Preventive Gerontology, Center for Gerontology and Social Science, National Center for Geriatrics and Gerontology, 7-430 Morioka-cho, Obu City, Aichi 474-8511, Japan.
Tel/Fax: +81-562-44-5651/+81-562-45-5638; E-mail: katayama.o@ncgg.go.jp

Round 2
Reviewer 1 Report
Thank you very much for the response, and for the modifications of the manuscript. In my opinion, the present form of the article is better, specially regarding the introduction and discussion sections. Congratullations for your work.